# Emergence agitation in pediatrics after dexmedetomidine vs. sevoflurane anesthesia: A randomized controlled trial

Corry Quando Yahya[1,2,3]*, Hori Hariyanto[1,2], Kohar Hari Santoso[3], Lucky Andriyanto[3], Arie Utariani[3], Bambang Pujo Semedi[3], Elizeus Hanindito[3], Yantoko Azis Priyadi[4]

**1** Department of Anesthesia and Intensive Care, Faculty of Medicine Universitas Pelita Harapan, Tangerang, Banten, Indonesia, **2** Department of Anesthesia and Intensive Care, Siloam Hospital Lippo Village, Tangerang, Banten, Indonesia, **3** Department of Anesthesiology and Reanimation, Faculty of Medicine Universitas Airlangga, Surabaya, Jawa Timur, Indonesia, **4** Department of Plastic and Reconstructive Surgery, Rumah Sakit Umum Persahabatan, Jakarta Timur, Indonesia

* corry.yahya@lecturer.uph.edu

## Abstract

### Introduction

Emergence agitation remains a problem that occurs in pediatric anesthesia. As cleft surgeries constitute one of the most common craniofacial surgeries encountered, majority of the children receive general anesthesia using high dose opioids and inhalation anesthetics and experience emergence agitation. Dexmedetomidine (DEX), an alpha-2 adrenoreceptor agonist possesses anxiolytic, sedative and analgetic properties and have been documented to reduce the incidence of postoperative agitation. Hence, this study aims to compare the incidence of emergence agitation between the use of intravenous DEX versus Sevoflurane (SEVO) anesthesia.

### Methods

This study selected one hundred twenty-one patients ages 3 months to 10 years with ASA 1 and 2 physical status scheduled to undergo elective cleft lip or cleft palate repair with general anesthesia. Before surgery, all patients were assessed preoperatively and subjects were divided into two groups using a computer-generated randomizer with 59 subjects selected as Dexmedetomidine group; and 62 subjects as Sevoflurane group. Extubation time, recovery time and emergence agitation scale were compared between the two groups.

### Results

This study found no significant difference in the extubation time between DEX and SEVO group (p = 0.317). The recovery time or time to attain full consciousness was statistically longer in the DEX group: 60 minutes as compared to 52

**Data availability statement:** The data will be made available from https://zenodo.org/records/16923848.

**Funding:** The author(s) received no specific funding for this work.

**Competing interests:** The authors have declared that no competing interests exist.

minutes in the SEVO group (p = 0.007). Emergence agitation assessed using Cravero score found that subjects from DEX group had an average Cravero score of 2.5; while SEVO group had an average Cravero score of 3.9 (p = < 0.001). The incidence of agitation was significantly higher in the SEVO group compared to the DEX group: 82% as compared to 10% (p = < 0.001) with an OR of 40.955 CI 95% (14.098–118.9).

## Conclusions

Dexmedetomidine significantly reduces the incidence of emergence agitation without prolonging extubation time in pediatric patients undergoing cleft lip and cleft palate surgery.

## Introduction

Emergence Agitation (EA) is a common occurrence in pediatric anesthesia. (1) The incidence has been reported to vary from 10% to as high as 80% after being exposed to inhalation anesthesia, particularly Sevoflurane. Maladaptive behaviors such as kicking, screaming, and thrashing are seen during agitation and contributes to postoperative complications such as tongue edema, re-bleeding of surgical wounds, falling out of bed, accidental removal of the intravenous line, bronchospasm and wound dehiscence. Anesthesia using Sevoflurane remains the main choice for cleft lip and cleft palate (CP) surgeries in children currently [1–3]. As a result, majority of these children experience intense agitation after recovering from anesthesia.

Dexmedetomidine has been used in clinical settings since 1999 and continues to gain popularity in the field of pediatric anesthesia. Unlike propofol and benzodiazepines, DEX does not act on gamma-aminobutyric acid (GABA) receptors [4,5]. Its hypnotic effects are mediated by the activation of central α2 presynaptic and postsynaptic receptors on the locus coeruleus, thereby resembling physiological deep sleep [6,7]. Anesthesia for pediatrics should ideally possess anxiolytic, sympatholytic and analgetic effects without the after effect of respiratory depression or post-anesthetic agitation.

Hence, this randomized controlled trial evaluated the use of total intravenous Dexmedetomidine compared to Sevoflurane inhalational anesthesia in children undergoing cleft lip and palate surgery. The primary endpoint is the incidence of emergence agitation while postoperative recovery profiles such as time to extubation and time to full recovery are the secondary endpoints.

## Materials and methods

This study was approved by the institutional ethics committee of Universitas Pelita Harapan, Faculty of Medicine and Siloam Hospital Lippo Village (183/K- LKJ/ETIK/V/2024, Date: 10 May 2024). The study is registered with the Protocol Registration and Results System, PRS. (ClinicalTrials.gov ID NCT06482125)

## Patients and Study design

This clinical trial was registered at https://register.clinicaltrials.gov (ClinicalTrials.gov ID NCT06482125) and conducted from July 31, 2024 to December 31, 2024. Written informed consent was obtained from the parents of each child.

A computer-generated random array sealed and envelope method was employed to divide enrolled pediatric patients into two groups by simple randomization (1:1 ratio allocation) which was performed on the anesthesia preoperative evaluation, one day before surgery. Experimental medication was prepared and provided by research staff who were not directly involved in patient care, while the surgeon, anesthesiologist and participating families were blinded to the medication distribution and group allocation. On the day of surgery, the anesthesiologist was informed on each child's allocation and gave medications according to their designated groups. Agitation scale upon awakening and extubation time were recorded by a nurse anesthetist inside the operating room; while agitation monitoring every fifteen minutes were carried out by a different nurse in the recovery room without prior knowledge of the child's group allocation.

Inclusion criteria were: American Society of Anesthesiologists (ASA) physical status I or II patients, weighing not more than 25 kg were scheduled to undergo cleft lip or palate repair. Exclusion criteria were children with congenital heart disease; children with syndromes such as Pierre Robin syndrome, Van der Woude syndrome, Stickler syndrome, Craniofacial dysmorphia, Trisomy 21, Trisomy 13, Velocardiofacial syndrome, Treacher Collins and Goldenhar syndrome (hemifacial microsomia), and/or children with a history of epilepsy and liver dysfunction.

## Surgical procedure

Preoperative assessment and randomization of group type was performed during anesthesia visitation, one day prior to the scheduled operation date. Anesthesia was administered using inhalation of 8% Sevoflurane in 100% oxygen until an intravenous line was secured. All patients were given Fentanyl 2 micrograms/kg, Propofol 3mg/kg and an oral endotracheal intubation of appropriate size was secured.

After intubation, patients in Group DEX stopped receiving Sevoflurane. Dexmedetomidine infusion using an intravenous line was started at a loading dose of 1.5ug/kg for 10 minutes, followed by a maintenance dose of 1.5ug/kg/hour via a syringe pump while patients in Group SEVO continued receiving Sevoflurane at a concentration of 2-3 vol%. All patients maintained spontaneous ventilation throughout the intraoperative period. Analgesia was supplemented with local anesthetic infiltration with Pehacaine® (Lidocaine 20mg/mL and Epinephrine 12.5 microgram/mL) without any adjuvants at the surgical site (lips or palate) and intravenous Paracetamol 15mg/kg given 10 minutes before the end of surgical procedure. At the end of surgery, Dexmedetomidine infusion was stopped in group DEX and Sevoflurane inhalation was stopped in group SEVO: every one of them was extubated and moved to the post-anesthesia care unit for postoperative monitoring.

## Data collection

All patients were admitted to Siloam Hospital Lippo Village and data were taken within the hospital vicinity: in the anesthesia preoperative clinic and operating theatres. Vital signs were monitored and recorded throughout the study. Standard monitoring includes electrocardiograph (ECG), non-invasive blood pressure and peripheral oxygen saturation ($SpO_2$) which were recorded starting from induction and every 5min, thereafter. Monitoring of anesthetic depth was ensured using SEDLine™ monitor, Masimo Corporation, Irvine, CA, USA to measure processed electroencephalograph or Patient State Index (PSI). Throughout the procedure, all patients retained their spontaneous respiration while maintaining their PSI value between 25-50.

The time of operation and awakening and any postoperative side effects such as laryngospasm, apnea, hypotension or bradycardia was recorded for both groups. Emergence agitation (EA) was assessed starting from extubation in the operating theatre and every 15 minutes in the post anesthesia care unit until the patients were awake, alert, calm and responsive to their parents. EA was assessed using the Cravero scale: (1) Obtunded with no response to stimulation (2) Asleep, but responsive

to movement or stimulation (3) Awake and responsive (4) Crying more than three minutes without any combative behavior (5) Agitated, thrashing behavior that requires restraint. The postoperative agitation scale of every child was monitored every 15 minutes until each child regains consciousness and evaluated as fit to be discharged to ward. Cravero agitation scores were summed and averaged for each patient recorded. Any score above 3.0 meant that the child experienced some agitation during their recovery period. Hence, scores greater than 3.0 were regarded as experiencing emergence delirium.

### Statistics analysis

The incidence of agitation among children anaesthetized with sevoflurane was estimated to be 40% [8] and we hypothesized that a lower incidence of 10% would be observed with dexmedetomidine. We determined that a sample size of 35 patients in each group would give us 80% power to detect this absolute 30% difference using statistical testing at the 5% level of significance. Adjusting our sample size of anticipated 20% dropouts, we aimed to recruit 42 patients per group.

Cravero agitation score between two groups were compared using Chi-square with continuity correction and Fischer exact test. Data were analyzed using SPSS software (SPSS 26.0, SPSS, Inc., Chicago, IL, USA). A p-value of less than 0.05 is considered as a significant difference.

### Results

A total of 130 children, ages 3 months to 10 years were found eligible, but nine were excluded due to the following reasons: 2 patients had Down syndrome; 3 patients were suspected of having Pierre Robin sequence and difficulty in airway management; 3 patients had congenital heart disease with ventricular septal defect or atrial septal defect and one patient with Tetralogy of Fallot. As a result, a total sample of 121 ASA I-II pediatric patients were randomly divided into group DEX (59 subjects) and group SEVO (62 subjects). Fig 1.

Prior to the surgical procedure, each subject was assessed for their preoperative ASA physical status. There were no significant differences in the basal characteristics of subjects such as age, body weight, duration of anesthesia and duration of operation. Duration of anesthesia averaged 48 minutes, while duration of surgery averaged 40 minutes. Table 1.

Hemodynamic parameters such as heart rate and blood pressure did not differ between both groups. Neither bradycardia nor hypotension were reported, but laryngospasm and desaturation occurred in both groups. Table 2. Even though surgical stimulus between labioplasty and palatoplasty differ, we did not observe any difference in the need for rescue opioids during intraoperative or postoperative period for both groups. No patients received additional opioids apart from their intubation dose.

After anesthesia, each child was awakened and extubation time recorded. Subjects in the DEX and SEVO group had an extubation time which are not statistically significant (p = 0.317). Longer time to awakening was noted for DEX group: 59.9 minutes compared to 51.7 minutes in the SEVO group (p = 0.007). Table 3, Fig 2.

After extubation, each child was transferred to the recovery room where agitation was assessed every 15 minutes until they regained full consciousness. Agitation scale was measured using a five-point Cravero scale, with its average score taken over a certain period of time. DEX group had a mean Cravero score of 2.5; while SEVO group had a mean score of 3.9 (p = <0.001). Fig 3.

Emergence agitation was defined as those with Cravero score greater than 3. In this study, emergence agitation was found in 82.3% (51/62) of SEVO group compared to 10.2% (6/59) of DEX group (p= <0.001). Further statistical analysis calculated an OR = 40.955, 95% CI, 14.098 -118.9, P=<0.001 between Sevoflurane anesthesia and the incidence of emergence agitation. Table 4, Fig 4.

### Discussion

Numerous studies have published Dexmedetomidine's efficacy in reducing emergence agitation in pediatrics [1,2,5,9–12]. Guler et al. found a significantly reduced incidence of emergence agitation in children undergoing adenotonsillectomy

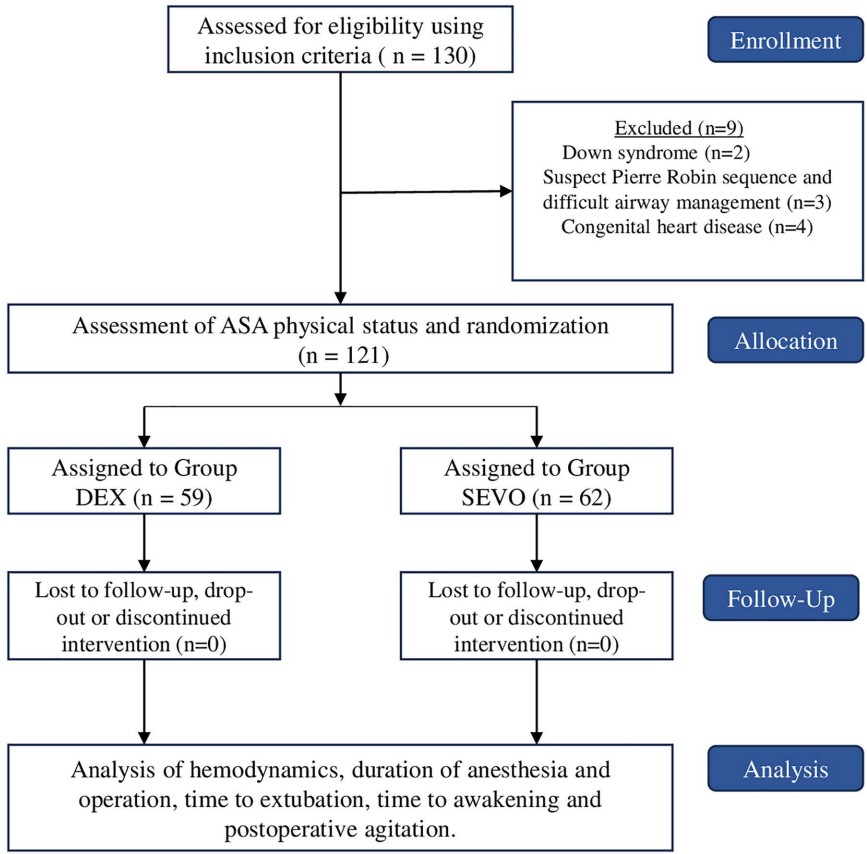

**Fig 1. Consort Flow Diagram.**

**Table 1. Demographic and surgical characteristics of patients.**

| Variable | Group S (n = 62) | Group D (n = 59) |
|---|---|---|
| | Range (Median) Mean ± SD | Range (Median) Mean±SD |
| Age (months) | 3-120 (24) | 3-120 (24) |
| | 34 ± 34 | 30 ± 28 |
| Weight (kg) | 5.2-25.6 (10) | 4.2-26.5 (9.4) |
| | 11.4 ± 5.4 | 10.6 ± 5.0 |
| Duration of surgery (min) | 41 ± 15 | 39 ± 14 |
| Duration of anesthesia (min) | 50 ± 17 | 46 ± 15 |
| Type of Operation (L/P) | 30/32 | 33/26 |

min = minutes, L = Labioplasty, P = Palatoplasty.

using a single administration of 0.5 ug/kg DEX five minutes before the end of operation [13]. A study on cleft lip and palate surgeries using different doses also reported significant reduction on emergence agitation compared to Sevoflurane anesthesia [8]. Recent meta-analysis and systematic review involving 7,714 children revealed consistent results on Dexmedetomidine's efficacy in reducing emergence agitation, no matter which route of administration was used: oral, intravenous,

**Table 2. Hemodynamic Parameter and Complications.**

| Variable | Group S (n=62) | Group D (n=59) |
|---|---|---|
| | Range (Median) | Range (Median) |
| | Mean±SD | Mean±SD |
| Heart rate (bpm) | 77-160 (117) | 82-139 (112) |
| | 118±19 | 110±15 |
| Systolic (mmHg) | 60-109 (84) | 75-111 (90) |
| | 83±11 | 92±9 |
| Diastolic (mmHg) | 30-65 (43) | 41-70 (50) |
| | 45±8 | 52±8 |
| **Complications** | **n (%)** | **n (%)** |
| Bronchospasm | 4 (6.5%) | 1 (1.7%) |
| Desaturation | 3 (4.8%) | 0 |

**Table 3. Postoperative Recovery Profiles.**

| Variable | Group S (n=62) | Group D (n=58) | P value | Mean Difference | SE Difference | CI 95% Difference |
|---|---|---|---|---|---|---|
| | Mean±SD | Mean±SD | | | | |
| Time To Extubation (min) | 9.5±3.7 | 10.4±4.6 | 0.317 | −0.841 | 0.754 | −2.334–0.652 |
| Time To Full Recovery (min) | 51.7±16.3 | 59.9±16.4 | 0.007 | −8.202 | 2.976 | −14.095–2.2310 |
| Cravero Score | 3.9±0.9 | 2.5±0.6 | < 0.001 | 1.4375 | 0.1331 | 1.1737–1.7013 |

*p<0.05 (significant), T-test.

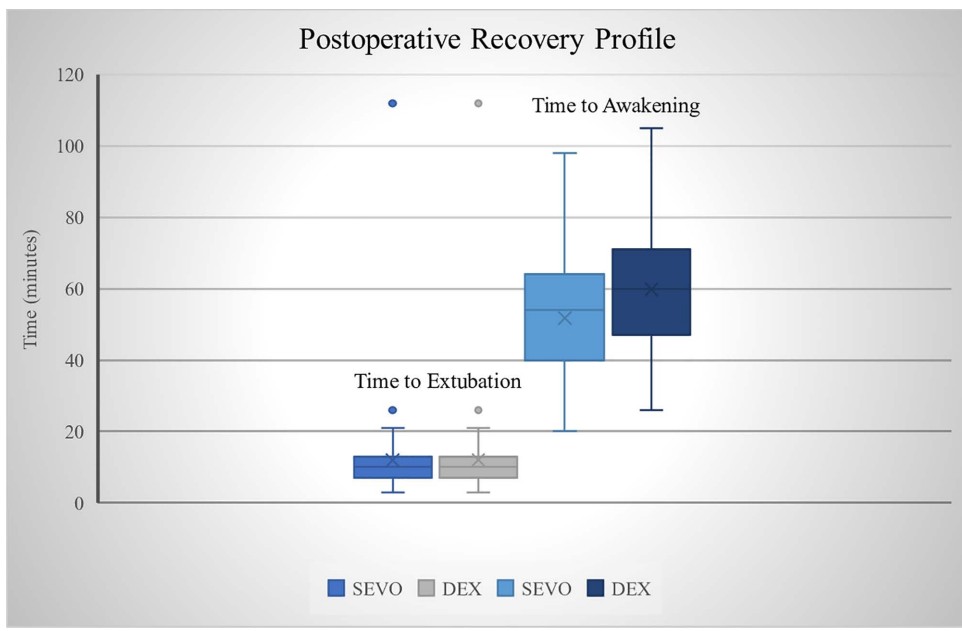

**Fig 2. Comparison of Sevoflurane and Dexmedetomidine recovery time.**

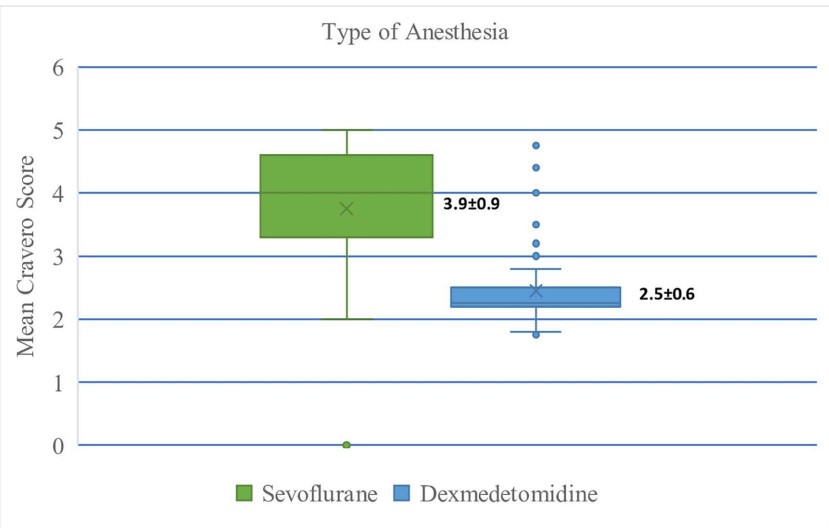

**Fig 3. Comparison of Sevoflurane and Dexmedetomidine mean Cravero score.**

**Table 4. Incidence of Emergence Agitation (Cravero Score > 3.0).**

| | Agitation | | | |
|---|---|---|---|---|
| **Group** | **Yes** | **No** | **p-value** | **OR (CI 95%)** |
| Sevo (n = 62) | 51 (82.3%) | 11 (17.7%) | <0.001 | 40.955 (14.098 - 118.972) |
| DEX (n = 59) | 6 (10.2%) | 52 (89.9%) | | |

*p < 0,05 (significant), Fischer exact test.

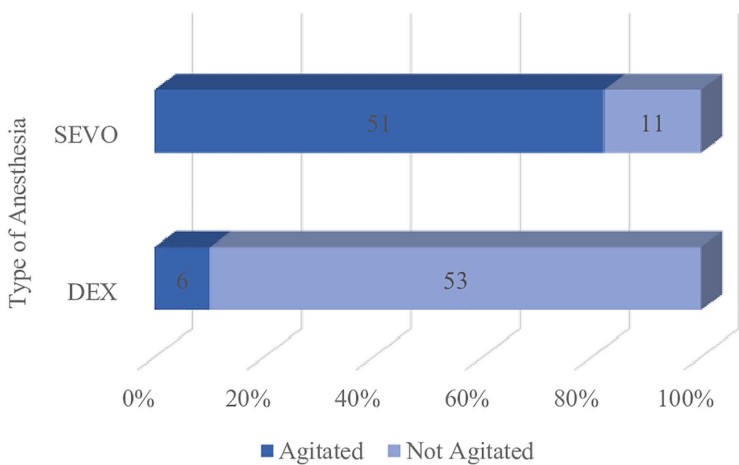

**Fig 4. Incidence of Emergence Agitation (Cravero score > 3.0) and Anesthesia type.**

intranasal, intramuscular or nebulized [14,15]. Dosages reported in literatures vary greatly with some using as low as 0.5ug/kg to as high as 5ug/kg along with different time, route and rate of administration [12].

Usual sedation dose ranges from 0.2 to 1.0 ug/kg/min; [1] however, dosages as high as 5 ug/kg using an intravenous route has been used in neonates to facilitate orotracheal repair without any adverse events [16–18]. Best evidence-based dosing recommends an intravenous loading dose of 0.5–2 µg/kg over 10 minutes followed by a maintenance dose of 0.5–1.5 µg/kg/h (1 month to 18 years of age) [1,19]. However, the dosage used in this study was derived from a preliminary study involving 23 pediatric patients undergoing cleft lip and palate repair with a loading dose of 1.5ug/kg administered within 10 minutes followed by 1.5ug/kg/hour continuous infusion [20].

In our study, all children were pre-induced with 8% Sevoflurane until venous access was established. Induction followed using Propofol 3mg/kg and Fentanyl 2µg/kg and airway secured. Subjects in SEVO group received 2-3% volume Sevoflurane under 50-50% O2/Air to maintain anesthesia; while children in DEX group received an initial loading dose continued by a maintenance dose. All children were able to maintain their spontaneous respiration throughout the procedure. Ensuring adequate anesthetic depth was of utmost importance, hence we utilized patient state index for both groups throughout the anesthetic procedure.

Reports on Patient State Index (PSI) compared to Bispectral Index (BIS) were reported to be equivalent in providing accurate anesthetic depth to children undergoing surgery ages 3-12 years [21–23]. However, some have found higher patient state index values in children less than two years of age, despite age-appropriate MAC values [24]. The reasons are unknown and there are no conclusive or gold standard tool to measure anesthetic depth of children less than two years, at the present moment. As such, this study utilizes Patient State Index to uniformly monitor and ensure a PSI value of 25-50 to all children; ranging from 3 months to 10 years of age.

Duration of anesthesia and surgery between both groups did not differ significantly, hence it can be concluded that all subjects received the same amount of treatment and exposure to each anesthetic agent. Hemodynamic derangements such as bradycardia and hypotension were reported as Dexmedetomidine's potential side effect [4]. However, our study did not encounter any adverse events during loading dose nor maintenance dose and a stable PSI level (25-50) was achievable throughout the intraoperative period. Extubation time were similar in both groups but time to attain full consciousness was slightly longer in the DEX group (52 minutes versus 60 minutes, p = 0.007).

Four cases of bronchospasm and three cases of desaturation were observed in group SEVO, while one case of bronchospasm was observed in group DEX. All of them resolved without the use of additional medication or adverse side effects (bradycardia, hypotension).

Postoperative pain remain one of the most common contributor to emergence agitation [25,26]. All patients in this study received the same analgetic regimen using local infiltration at the incision site and intravenous Paracetamol at 15mg/kg. The use of local infiltration contributed to reduced pain stimulus for both DEX and SEVO groups, resulting in no administration of rescue opioids during the intraoperative and postoperative period. Despite this, majority of patients (82%) from the SEVO group still experience intense agitation with hysterical crying, kicking, pulling, arching their back and frantic movement of their extremities compared to 10% of patients from the DEX group. Perhaps there might be another mechanism to the cause of emergence agitation as non-painful procedures such as magnetic resonance imaging sedation under sevoflurane anesthesia may still produce emergence agitation [27]. Periodic bursts of neural activity and differential recovery rate have been elucidated as the cause of hyperactivity behavior seen during Sevoflurane agitation [28]; but the reason as to why some children in the DEX group experience emergence agitation remain unanswered.

Overall, we found children exposed to Sevoflurane anesthesia had a higher incidence of emergence agitation compared to those receiving Dexmedetomidine anesthesia. The results are consistent with similar studies conducted by Peng and Zhang which found 90% agitation in SEVO group vs 15% agitation in the placebo group to children undergoing cleft and lip palate surgery [8]. Another study conducted by Guler et al on children undergoing adenotonsillectomy found 57% incidence of agitation in the SEVO group compared to 15% in the DEX group [13].

Comparing studies is difficult because of heterogeneity between population, surgical site involved, assessment tool used and different protocols implemented thereby producing variable results. Nevertheless, results on DEX have consistently found a reduction in the incidence of emergence agitation to children undergoing anesthesia.

Our study demonstrated that DEX can be used as a single anesthetic agent to maintain adequate hypnosis using a targeted Patient State Index (PSI) of 25-50 without any adjuvants, limiting bias of other anesthetic contribution to patient outcome. However, limitation of the current study was the short duration of surgical time and revised definition of emergence delirium. It should be noted that the surgical technique for palatoplasty employed in this study is the Kilner V-Y Push Back Method, with an average time confined to less than one hour. Procedures greater than one hour under DEX anesthesia may produce different time to awakening results and hence warrants further study. In this study, emergence delirium was defined as Cravero scale greater than three and thus differ from the established definition.

## Conclusion

In summary, a 1.5 µg/kg (for 10 minutes) loading dose of dexmedetomidine followed by a maintenance infusion dose of 1.5 µg/kg/hour is safe and reduces the incidence of emergence agitation in children undergoing cleft lip and palate surgery.

## Supporting information

**S1 Data. Study Protocol English.**
(PDF)

**S2 Data. Consort Checklist updated.**
(DOC)

**S3 Data. Protokol PLOS ONE Indo.**
(PDF)

**S4 Data. SPSS Data to excel 121.**
(XLSX)

## Author contributions

**Conceptualization:** Corry Quando Yahya, Hori Hariyanto, Kohar Hari Santoso, Lucky Andriyanto, Arie Utariani, Bambang Pujo Semedi, Elizeus Hanindito, Yantoko Azis Priyadi.

**Data curation:** Corry Quando Yahya, Hori Hariyanto, Lucky Andriyanto, Arie Utariani, Bambang Pujo Semedi, Elizeus Hanindito, Yantoko Azis Priyadi.

**Formal analysis:** Corry Quando Yahya, Hori Hariyanto, Kohar Hari Santoso, Lucky Andriyanto, Arie Utariani, Bambang Pujo Semedi, Elizeus Hanindito, Yantoko Azis Priyadi.

**Investigation:** Corry Quando Yahya, Hori Hariyanto, Kohar Hari Santoso, Lucky Andriyanto, Arie Utariani, Elizeus Hanindito.

**Methodology:** Corry Quando Yahya, Hori Hariyanto, Kohar Hari Santoso, Lucky Andriyanto, Bambang Pujo Semedi, Elizeus Hanindito, Yantoko Azis Priyadi.

**Project administration:** Corry Quando Yahya, Hori Hariyanto, Yantoko Azis Priyadi.

**Resources:** Corry Quando Yahya, Hori Hariyanto, Elizeus Hanindito, Yantoko Azis Priyadi.

**Software:** Corry Quando Yahya.

**Supervision:** Corry Quando Yahya, Kohar Hari Santoso, Lucky Andriyanto, Arie Utariani, Bambang Pujo Semedi, Elizeus Hanindito.

**Validation:** Corry Quando Yahya, Hori Hariyanto, Kohar Hari Santoso, Lucky Andriyanto, Arie Utariani, Bambang Pujo Semedi, Elizeus Hanindito.

**Visualization:** Corry Quando Yahya, Hori Hariyanto, Lucky Andriyanto, Arie Utariani, Bambang Pujo Semedi, Elizeus Hanindito.

**Writing – original draft:** Corry Quando Yahya, Hori Hariyanto, Kohar Hari Santoso, Lucky Andriyanto, Arie Utariani, Bambang Pujo Semedi, Elizeus Hanindito.

**Writing – review & editing:** Corry Quando Yahya, Hori Hariyanto, Kohar Hari Santoso, Lucky Andriyanto, Arie Utariani, Bambang Pujo Semedi, Elizeus Hanindito.

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
