## [Decision Letter · Decision Letter 0]

17 May 2025

PONE-D-25-16267Emergence Agitation in Pediatrics after Dexmedetomidine vs. Sevoflurane Anesthesia: a randomized controlled trialPLOS ONE

Dear Dr. Yahya,

Thank you for submitting your manuscript to PLOS ONE. After careful consideration, we feel that it has merit but does not fully meet PLOS ONE’s publication criteria as it currently stands. Therefore, we invite you to submit a revised version of the manuscript that addresses the points raised during the review process.

**The reviewers have made specific comments, I am looking forward to the revised version of manuscript**

We look forward to receiving your revised manuscript.

Kind regards,

Benjamin Benzon, Ph.D., M.D.

Academic Editor

PLOS ONE

Journal Requirements:

2. Please include a caption for figure 1, 2, 3.

3. Please remove all personal information, ensure that the data shared are in accordance with participant consent, and re-upload a fully anonymized data set.

Reviewers' comments:

Reviewer's Responses to Questions

**Comments to the Author**

1. Is the manuscript technically sound, and do the data support the conclusions?

Reviewer #1: Yes

Reviewer #2: Partly

2. Has the statistical analysis been performed appropriately and rigorously? 

Reviewer #1: Yes

Reviewer #2: No

3. Have the authors made all data underlying the findings in their manuscript fully available?

Reviewer #1: No

Reviewer #2: Yes

4. Is the manuscript presented in an intelligible fashion and written in standard English?

Reviewer #1: No

Reviewer #2: Yes

5. Review Comments to the Author

Reviewer #1: Thank you for allowing me to read this manuscript. I congratulate Dr. Yahya and colleagues with their very interesting study on the effects of dexmedetomidine and sevoflurane on emergency agitation following pediatric anesthesia. The topic is of great interest given the high incidence of emergency agitation and its consequences which are very well described in the manuscript. The study is timely as dexmedetomidine is an emerging drug that has garnered increased attention, especially in pediatric anesthesia. There are several issues that will need to be addressed to find this manuscript acceptable for publication.

1. I was surprised to read that dexmedetomidine was used as the sole anesthetic agent. To my knowledge, it is primarily used as an adjuvant to propofol or sevoflurane based maintenance of anesthesia, or as a sedative for procedures that require less depth of anesthesia. I can imagine that for labioplasty, local infiltration combined with dexmedetomidine sedation may be sufficient, however I find it hard to believe that a palatoplasty can be performed with only dexmedetomidine (i.e. without other anesthetics). The authors state that the patient state index (PSI) was always between 25-50. Was this achieved without any dose titration and was it measured in all cases continuously? Please show this data to support this claim as to be honest I find it highly unlikely. Is there any literature on dexmed as sole anesthetic and PSI values? Also please provide evidence that PSI is a reliable tool to assess anesthetic depth for this anesthetic and this age group.

2. For this reason, please also provide a subanalysis for labioplasty vs palatoplasty as the surgical stimulus is distinctly different.

3. Also, please specify more details on the surgical procedure. In my experience, a good palatoplasty takes longer than the times reported in the manuscript. What did the surgeons actually do? This is relevant as it relates to required anesthetic depth.

4. Were any opioids used intraoperatively in addition to the intubation dose? Please provide this data per allocated group (% of patients with opioids, mean dose)

5. Emergency delirium is defined as Cravery >= 4 for > 5 mins. Why did the authors define it as Cravery >= 3 instead of 4?

6. Please specify the reasons for the 9 exclusions.

7. What local infiltration is used? Please specify drug, additives and concentration. Infiltrated prior to surgical procedure or afterwards?

8. The authors state a notably short time to recovery after dexmedetomidine. In clinical practice dexmedetomidine has quite a long half life. What do the authors mean with “full recovery”? Were these children really awake and ready for discharge, or were they arousable? Potentially there was little arousal in the dexmed group as they were simply still sedated?

9. Please specify the incidence of bradycardie, which is a well known side effect of dexmedetomidine.

10. Did the authors consider using a suprazygomatic maxillary nerve block, which is known to provide excellent postoperative analgesia for palatoplasty, therefore facilitating a smooth recovery?

11. The sample size calculation requires 84 patients however the report is on 121 cases. Please give a justification for 45% more patient inclusions. It is questionable to subject more participants than necessary to a research project, especially if one of the two interventions turns out to be superior.

12. The authors report that the data is fully available however I was unable to find it. Please give more instructions on where to find these data.

13. Lastly, this manuscript would benefit from extensive language editing.

Reviewer #2: Abstract

- line 38: "majority of the children receive" not "receives"

- line 43: better to say "intravenous DEX versus sevoflurane" otherwise the 'and' implies treatment with a combination of DEX and SEVO.

- results should include mean differences or odds/risk ratios for all outcomes, with 95% confidence intervals in addition to p-values, not p-values only. The primary outcome should be reported first. Small p-values should be reported as p < 0.001 not p = 0.000 which implies a zero p-value, which is incorrect.

Introduction

- line 74: "remains" rather than "remain". The end of that sentence could be shortened to "currently" instead of "at the current moment".

Methods

- all mentions of the numbers of children randomised (e.g. line 113) are results and should be reported in the results section

- the sample size calculation is poorly reported. I would suggest: "The incidence of agitation among children anaesthetised with sevoflurane was estimated to be 40% [ref] and we hypothesised that a lower incidence of 10% would be observed with dexmedetomidine. We determined that a sample size of 35 patients in each group would give us 80% power to detect this absolute 30% difference using statistical testing at the 5% level of significance. Adjusting our sample size of anticipated 20% dropouts, we aimed to recruit 42 patients per group."

- the statistical analysis is poorly reported. It is not clear exactly what was analysed by Kolmogrov-Smirnov (line 174). Generally, there is no role for these tests in a clinical trial analysis. It is also not clear what analysis of age and weight (line 175) were required, as none of these are outcomes. Further, there is no role for any of the tests listed in line 177 to 179. This being a straightforward trial analysis, what were required were (1) a descriptive analysis of participant characteristics in each of the two groups according to baseline characteristics, with no statistical tests comparing the groups (this had been done in Table 1, although all reference to statistical tests should be removed, with no other changes), and (2) an analysis comparing outcomes between the groups, with mean (SE) for continuous outcomes or count (%) for binary outcomes in each arm, followed by unadjusted effects (mean difference, risk ratio, odds ratio etc) with 95% CI and p-value, and adjusted effects with 95%CI and p-value. If the parameters reported in Table 2 are considered outcomes, they should also be reported in this format, preferably with the other outcomes

Results

- in the CONSORT diagram, reasons for exclusion should be reported

- throughout the results effects should be reported as mean differences or odds/risk ratios for all outcomes, with 95% confidence intervals in addition to p-values, not p-values only

- see comments above regarding tables 1 to 3.

6. PLOS authors have the option to publish the peer review history of their article (what does this mean? ). If published, this will include your full peer review and any attached files.

**Do you want your identity to be public for this peer review?** For information about this choice, including consent withdrawal, please see our Privacy Policy .

Reviewer #1: No

Reviewer #2: No

---

## [Author Response · Author response to Decision Letter 1]

21 Jun 2025

Dear Reviewer 1 and Reviewer 2,

We would like to thankyou for your comments and suggestions to our manuscript. We have made the necessary corrections and comments that you have provided. Attached with our submission is our reply to your review answered individually. We thankyou and appreciate your time in reviewing our manuscript.

Sincerely,

Corry Q Yahya et al.

---

## [Decision Letter · Decision Letter 1]

30 Jul 2025

PONE-D-25-16267R1Emergence Agitation in Pediatrics after Dexmedetomidine vs. Sevoflurane Anesthesia: a randomized controlled trialPLOS ONE

Dear Dr. Yahya,

Thank you for submitting your manuscript to PLOS ONE. After careful consideration, we feel that it has merit but does not fully meet PLOS ONE’s publication criteria as it currently stands. Therefore, we invite you to submit a revised version of the manuscript that addresses the points raised during the review process.

We look forward to receiving your revised manuscript.

Kind regards,

Benjamin Benzon, Ph.D., M.D.

Academic Editor

PLOS ONE

Journal Requirements:

**Additional Editor Comments:**

There are still some minor issues in my opinion that should be addressed (such as 95% CI of effect mentioned by the Reviewer 2), Reviewer 1 has made some pretty reasonable comments also, please address all of them and I am looking forward to receive the revised version of manuscript.

Reviewers' comments:

Reviewer's Responses to Questions

**Comments to the Author**

1. If the authors have adequately addressed your comments raised in a previous round of review and you feel that this manuscript is now acceptable for publication, you may indicate that here to bypass the “Comments to the Author” section, enter your conflict of interest statement in the “Confidential to Editor” section, and submit your "Accept" recommendation.

Reviewer #1: (No Response)

Reviewer #2: (No Response)

2. Is the manuscript technically sound, and do the data support the conclusions?

Reviewer #1: Partly

Reviewer #2: Partly

3. Has the statistical analysis been performed appropriately and rigorously? 

Reviewer #1: Yes

Reviewer #2: No

4. Have the authors made all data underlying the findings in their manuscript fully available?

Reviewer #1: Yes

Reviewer #2: Yes

5. Is the manuscript presented in an intelligible fashion and written in standard English?

Reviewer #1: Yes

Reviewer #2: Yes

6. Review Comments to the Author

***Reviewer #1***: A1. Thank you for your answer. Please include the video file in the supplementary files. Also, please discuss PSI and its limitations in patients < 3 yr and in the setting of DEX, in the Discussion.

2. Please report this subanalysis in the Results section and discuss its relevance in the Discussion.

A3. Please discuss in the Discussion that the results are only generalizable to the Kilner V-Y Push Back method. These results should not be confused with the more elaborate and lengthier procedures commonly performed in for example European centers for palatoplasty.

A4. Please mention this in the Results section.

A5. Please include in the Limitations that you were not able to accurately use the definition of emergence delirium (Cravero >= 4 for > 5 mins) due to the way in which data was collected.

A6. You could underline "Excluded (n=9)" to emphasize that the other lines outline the reasons for exclusion

A7. Please include this in the Methods section, including whether or not dexamethasone is given (this can prolong the block)

A10. Please include this in the Discussion to allow comprehensive interpretation of the data.

A11. I disagree. By including more patients than necessary, you subjected more than necessary subjects to an intervention which you later show to be inferior. If you really believe more patients is better, why did you stop at 121 cases? You should specifically report that recruitment of subjects was continued after reaching the aimed sample size and provide a reason. Simply stating that larger studies have more statistical power, is insufficient.

A12. Thank you

A13. Thank you

***Reviewer #2:*** The authors have made some changes which have improved this manuscript. However, they have not responded to (or explained why the disagree with) my comments from the previous round.

For example, I had recommended that the abstract include effects with 95% confidence intervals in addition to the p-value; these have not been included and no explanation or refutation of my suggestion have been provided.

Additionally, for the statistical analysis, I had indicated that the approach described was unnecessary for a trial, that instead the analysis should include: (1) a descriptive analysis of participant characteristics in each of the two groups according to baseline characteristics, with no statistical tests comparing the groups, and (2) an analysis comparing outcomes between the groups, with mean (SE) for continuous outcomes or count (%) for binary outcomes in each arm, followed by unadjusted effects (mean difference, risk ratio, odds ratio etc) with 95% CI and p-value, and adjusted effects with 95%CI and p-value. However, the methods still mention Kolmogrov-Smirnov testing, Mann-Whitney tests, student's t-tests and chi-squared tests. Granted, the unnecessary group comparisons that were included in Table 1 have now been removed, however, the results in Table 3 do not still include estimates of effects with 95% confidence intervals.

Lastly, there are still some results included in the methods, for example, the statement about the number of participants included in line 176 and figure 1 (the CONSORT diagram) - these are results and should be reported in the results section.

I recommend that the authors look at the CONSORT statement for guidance on how to report the results of a trial, and to consult a statistician who is experienced with analysis and reporting of trials when revising the manuscript.

7. PLOS authors have the option to publish the peer review history of their article (what does this mean? ). If published, this will include your full peer review and any attached files.

**Do you want your identity to be public for this peer review?** For information about this choice, including consent withdrawal, please see our Privacy Policy .

Reviewer #1: No

Reviewer #2: No

---

## [Author Response · Author response to Decision Letter 2]

21 Aug 2025

Response to Reviewer #1 and Reviewer #2 are attached as word files. Please find them in the attached files.

---

## [Decision Letter · Decision Letter 2]

17 Sep 2025

Emergence Agitation in Pediatrics after Dexmedetomidine vs. Sevoflurane Anesthesia: a randomized controlled trial

PONE-D-25-16267R2

Dear Dr. Yahya,

We’re pleased to inform you that your manuscript has been judged scientifically suitable for publication and will be formally accepted for publication once it meets all outstanding technical requirements.

Kind regards,

Chiara Lazzeri

Academic Editor

PLOS ONE

Additional Editor Comments (optional):

Reviewer #1:

Reviewer #2:

Reviewers' comments:

Reviewer's Responses to Questions

**Comments to the Author**

1. If the authors have adequately addressed your comments raised in a previous round of review and you feel that this manuscript is now acceptable for publication, you may indicate that here to bypass the “Comments to the Author” section, enter your conflict of interest statement in the “Confidential to Editor” section, and submit your "Accept" recommendation.

Reviewer #1: (No Response)

Reviewer #2: (No Response)

2. Is the manuscript technically sound, and do the data support the conclusions?

Reviewer #1: Yes

Reviewer #2: Partly

3. Has the statistical analysis been performed appropriately and rigorously? 

Reviewer #1: Yes

Reviewer #2: No

4. Have the authors made all data underlying the findings in their manuscript fully available?

Reviewer #1: Yes

Reviewer #2: Yes

5. Is the manuscript presented in an intelligible fashion and written in standard English?

Reviewer #1: Yes

Reviewer #2: Yes

6. Review Comments to the Author

Reviewer #1: Thank you very much for the improvements to the manuscript. I am impressed and believe it is now close to being suitable for publication. I would, however, recommend clarifying in your Results section that recruitment and inclusion of patients continued for the entire pre-specified study duration, even though the planned sample size had already been reached before that period ended. Please include both parts of that sentence.

Reviewer #2: In my previous comments I had recommended that the authors report effects as mean differences or odds ratios or risk ratios with 95% confidence intervals and p-values, on the assumption that they would conduct regression analysis of outcomes. This has not been done for some outcomes, and I can see and understand why. I would have recommended regression analysis throughout, however, the current analysis would be acceptable with some further changes.

1. in the abstract where the authors say there was no difference in extubation time, it would be informative to report the mean (with standard error) extubation times in each group before the p-value for the difference. In the next sentence mentioning recovery times, the means have been reported, but the standard errors should also be reported for each mean.

2. it is unusual to report mean Cravero scores and yet analyse it as a categorical variable (using chi-squared tests). This scale seems to me to be an ordered categorical scale. If you want to report some average, perhaps report the median or mode of the score - I suspect that the average you have reported is an arithmetic mean, which doesn't seem suitable for an ordered scale. However, if this is the standard practice for reporting this scale then it is fine (although strictly speaking not appropriate).

3. it is helpful that when you report outcomes, you should include as much detail as possible. For example, where you report the incidence of agitation in the two groups, you need to include the number of agitation events and denominator in each group, i.e. x/y events in group a (a prevalence of m%) versus u/v events in group b (a prevalence of n%), relative risk or odds ratio x.xx, 95%CI x.xx to x.xx, p-value x.xxx. Having now read through to line 257, this is present in the main text but should also be in the abstract.

4. I suggest re-wording the statistical analysis from line 178 to line 182 as follows: "We examined the distributions of continuous outcomes using the Kolmogrov-Smirnov test and subsequently compared normally-distributed outcomes using student t-tests and non-normal outcomes using the Mann-Whitney test". You do not need to say which outcomes were normal/not normal, as that statement is a result and would need to be in the results section (although you don't really need to say it as it will be obvious when one reads the results). That aside, there is generally no role for tests of normality in such analyses, as they are sensitive to small deviations from normality, hence the standard practice is to simply use regression analyses for all continuous measures (not ordinal ones). Additionally, there is no need to test normality for baseline characteristics such as age and weight.

5. For line 183, consider my suggestion regarding the ordinal nature of this scale, and the appropriate 'average' to report should you choose to do so. If you report the median please report it with the interquartile range.

6. you do not need to repeat that "Agitation scale was measured using a five-point Cravero scale" in line 232 as this has already been described in the methods. See my comment in 5 above about reporting the 'average' Cravero score in each arm

7. indicate whether the quantities reported for time to awakening are means or medians - if they are means, also report the standard errors in addition to the p-values e.g. "Longer time to awakening was noted for DEX group: mean (SE) 59.9 (xx.x) minutes compared to 51.7 (xx.x) minutes in the SEVO group (p = 0.007)."

8. Throughout where I have suggested adding standard errors I mean standard errors and not standard deviations; where you have reported standard deviations in Tables 1 and 2, this is fine as these are descriptive features and not outcomes; however in table 3, as these are outcomes, you should replace the standard deviations with standard errors. When you do this go through the text and ensure it reflects these changes throughout.

9. Figure 3 aligns with my thinking regarding the Cravero score - the average you should be reporting is the median, which the box-and-whisker plots show well, although it is okay to have the mean too in case this is what is typically reported for this scale. I would suggest a single change to this figure, which is to remove the word 'Mean' from the title of the y-axis, and add a legend to say that the x indicates the mean. The median is self-explanatory from the figure. I would also remove the SDs from this figure.

10. it is only when I get to line 257 that I realise how you defined emergence agitation from the Cravero score; this should have been in the methods.

7. PLOS authors have the option to publish the peer review history of their article (what does this mean? ). If published, this will include your full peer review and any attached files.

**Do you want your identity to be public for this peer review?** For information about this choice, including consent withdrawal, please see our Privacy Policy .

Reviewer #1: No

Reviewer #2: No

---

## [Editor Report · Acceptance letter]

PONE-D-25-16267R2

PLOS ONE

Dear Dr. Yahya,

I'm pleased to inform you that your manuscript has been deemed suitable for publication in PLOS ONE. Congratulations! Your manuscript is now being handed over to our production team.

Kind regards,

on behalf of

Dr. Chiara Lazzeri

Academic Editor

PLOS ONE